# The Vaginal Microbiome in Health and Disease—What Role Do Common Intimate Hygiene Practices Play?

**DOI:** 10.3390/microorganisms11020298

**Published:** 2023-01-23

**Authors:** Alexandra M. Holdcroft, Demelza J. Ireland, Matthew S. Payne

**Affiliations:** 1School of Biomedical Sciences, University of Western Australia, Crawley, WA 6009, Australia; 2Division of Obstetrics and Gynaecology, University of Western Australia, Subiaco, WA 6008, Australia

**Keywords:** vaginal microbiome, vagina, female hygiene, sexual health

## Abstract

The vaginal microbiome is a dynamic, sensitive microenvironment. The hallmark of a ‘healthy’ vaginal microbiome is currently believed to be one dominated by *Lactobacillus* spp., which acidifies the vaginal environment and help to protect against invading pathogens. However, a ‘normal’ microbiome is often difficult, if not impossible, to characterise given that it varies in response to numerous variables, including pregnancy, the menstrual cycle, contraceptive use, diet, ethnicity, and stress. A *Lactobacillus*-depleted microbiome has been linked to a variety of adverse vaginal health outcomes, including preterm birth (PTB), bacterial vaginosis (BV), and increased risk of sexually transmitted infections. The latter two of these have also been associated with feminine intimate hygiene practices, many of which are practised without any evidence of health benefits. The most extensively studied practice is vaginal douching, which is known to cause vaginal dysbiosis, predisposing women to BV, pelvic inflammatory disease, and PTB. However, little is known of the impact that intimate hygiene practices and associated products have on the vaginal microbiome. This review aims to outline the major factors influencing the vaginal microbiome and common vaginal infections, as well as to summarise current research surrounding the impact of hygiene products and practices on the vaginal microbiome.

## 1. Introduction

The vaginal microbiome is a dynamic, sensitive microenvironment that changes in response to pregnancy, the menstrual cycle, contraceptive use, and diet [1]. The vaginal microbiota lives in a mutualistic relationship with the host, providing protection from pathogenic bacteria in exchange for nutrients and shelter [1,2,3]. A significant amount of protection is provided by bacteria from the genus *Lactobacillus* spp., which produces lactic acid that contributes to the acidic vaginal pH [1,3]. A normal, healthy microbiota is dominated by a variety of *Lactobacillus* spp., including *L. crispatus, L. gasseri, L. iners* and *L. jensenii* [1,4]. Protective *Lactobacillus* titres can be easily disrupted, resulting in vaginal dysbiosis and predisposing women to a variety of adverse vaginal health conditions such as bacterial vaginosis (BV), candidiasis (thrush), and sexually transmitted infections (STIs) [1,2,3].

The term ‘feminine hygiene habits’ encompasses a wide variety of practices used to cleanse in and/or around the female genital area. The most well-studied vaginal hygiene practice, douching, involves the introduction of water and/or cleansing products into the vagina. Douching has been associated with increased risks of BV, preterm birth, and pelvic inflammatory disease (PID), which can lead to infertility [5,6]. It has been hypothesised that douching alters the microbial community within the vagina, causing inflammation and providing an opportunity for pathogenic bacteria to invade and colonise the area [7]. Aside from vaginal douching, little is known of the effect that other feminine hygiene products such as gels, sprays and wipes have on the vaginal microbiome.

This review aims to provide an overview of the major factors influencing the vaginal microbiome, along with a critical appraisal of the literature pertaining to feminine hygiene products and practices and their associated impact. Current literature frequently uses the term ‘vagina’ as all-encompassing for the genital area, placing uncertainty on whether various feminine hygiene products are being used internally or externally; this is a major limitation of previous studies.

## 2. Feminine Hygiene Practices

The vagina is a self-cleansing organ [8]. Vaginal discharge consists of desquamated vaginal epithelial cells, bacteria, and glandular secretions and helps to protect against vulvovaginal infections [7]. Discharge is normally white or clear in colour and possesses a slight, non-offensive odour. The quality of discharge changes throughout the menstrual cycle: discharge is thick, sticky, and hostile to sperm early in the cycle, and becomes thin and watery during ovulation due to rising estrogen levels [7]. For some women, discharge is unpleasant, and this leads to the use of feminine hygiene products and practices to remove discharge and odour from their genital area.

In some populations, up to 95% of women have used at least one feminine hygiene product or practice in or around their genital area [9], despite little being known of their short and long-term health effects. The type and frequency of use of these practices varies and may be related to personal preference or societal, cultural, and religious influences [7,10]. Aside from vaginal douching, women also use products specifically for the intimate area including vaginal washes, wipes, and sprays. Some women also report the use of general cleansing products such as baby wipes, oils, and moisturisers [9].

Feminine hygiene products such as douches, wipes, sprays, washes, and powders are part of a fast-growing industry worth USD 2 billion in the US [11]. These products are marketed to women with the idea of maintaining a ‘clean and fresh’ vagina. Products such as ‘Femfresh’ and ‘Vagisil’ are scented to block vaginal odour which in most cases is completely normal and healthy. These marketing ploys capitalise on cultural messages that women’s bodies are problematic, unclean, and require cosmetic products to maintain a healthy state [10].

Research suggests that the use of feminine hygiene products may be the result of a ‘harmful cycle’ whereby women wash to reduce perceived itching, odour, and discharge, only to develop more significant or additional symptoms resulting from increased washing and the associated disturbance of the normal microbiome [9,12]. By increasing the knowledge surrounding feminine hygiene products, the vaginal microbiome, and adverse vaginal health conditions, women will be able to make an informed choice about their use of these products to optimise their reproductive health.

## 3. The Vaginal Microbiome

The vaginal microbiome is a dynamic ecosystem that varies between women, depending on several factors. A seminal study by Ravel et al. [4] introduced the idea of community state types (CST) after discovering that the microbiomes of women of varying ethnicities could be clustered into five core community groups. Four of these groups were dominated by *Lactobacillus* spp.*, (L. crispatus,* CST I; *L. gasseri*, CST II; *L. iners*, CST III; *L. jensenii*, CST V) while the final group (CST IV) was characterised by a low relative abundance of *Lactobacillus* spp. with higher proportions of anaerobic bacteria [4]. In 2012, Gajer et al. divided CST IV into two sub-states, with CST IV-A dominated by anaerobes of the genera *Anaerococcus* sp., *Prevotella* sp. and *Streptococcus* sp., and CST IV-B by higher proportions of the genera *Atopobium* sp. and *Megasphaera* sp., amongst others [13].

It has since been recognised that a healthy vaginal microbiome can be dominated by *Bifidobacterium* sp. [14,15,16,17]. *Bifidobacterium* sp. are a group of Gram-positive, anaerobes that are known to colonise the human vagina, oral cavity, and the gastrointestinal tract (GIT) where they play an important role in the protection from pathogens through the production of bacteriocins [14]. At present, vaginal Bifidobacteria are poorly characterised due to the limited coverage of some 16S rRNA gene primer sets and as a result, are often missed in many studies. In particular, the primer pair used by Ravel et al. [4] had only 12.9% coverage of *Bifidobacterium* sp. sequences within the SILVA database [18]. However, a study by Freitas et al. [14] used quantitative PCR to confirm the relative abundance of *Bifidobacterium* sp. in the vaginal microbiomes of 42 healthy reproductive-aged women. Interestingly, they found that 4.2% of these women had *Bifidobacterium*-dominant profiles. They also reported that *Bifidobacterium* were very capable of producing lactic acid, and could tolerate a low pH, which is typical of healthy vaginal fluid. This suggests that *Bifidobacterium* spp. may be as protective as *Lactobacillus* spp. in preventing vaginal colonisation by pathogenic organisms [14].

## 4. Factors Influencing the Normal Vaginal Microbiome

A ‘normal’ vaginal microbiome is difficult, if not impossible, to define. CST I, II, III and V have all been considered ‘healthy’ for their dominance of *Lactobacillus* species (*L. crispatus, L. gasseri, L. iners, L. jensenii*). However, as living microcosms, microbiomes are subject to change in response to intrinsic factors such as menstrual cycling and pregnancy but also external factors such as diet, exposure to smoke and other airborne pollutants, antibiotic treatment, exercise, and stress.

### 4.1. Ethnicity

Studies on North American, Japanese, and Chinese women have reported that microbiomes dominated by one or more *Lactobacillus* spp. were most prevalent [15,19,20,21,22]. Fettweis et al. added European women to the *Lactobacillus*-dominant microbiome group, whilst showing African American women exhibited diverse microbial profiles with reduced concentrations of Lactobacilli [19]. In fact, they discovered that the *Lactobacillus*-depleted CST IV profile was four times more common in Black women than Caucasian women. These findings were also consistent with Zhou et al. [20], who reported that Black women were less likely to possess a vaginal microbiome dominated by Lactobacilli compared with Caucasian women. Ravel et al. [4] enabled a more granular assessment of the impacts of ethnicity on the microbiome. 396 North American women of varying ethnicities (White, Black, Hispanic, and Asian) were studied with statistically significant differences in the proportions of each CST reported among the four ethnicities [4]. Specifically, Asian and White women were more likely to have *Lactobacillus*-dominant vaginal bacterial communities such as CST I, II, III, and V than Black or Hispanic women. Additionally, CST IV, which is dominated by anaerobes, was overrepresented in Black and Hispanic women; however, it is unclear what proportion of the Hispanic cohort were Black and White Hispanic, which may have influenced the results observed. These findings suggest that the vaginal microbiome may be genetically determined, however, given that diet and hygiene practices also differ according to culture and ethnicity, it is likely to be influenced by a wide variety of factors.

### 4.2. Diet

Research on the gut microbiome has revealed the effect of diet on gut bacterial composition, which impacts the well-being of individuals and their susceptibility to diseases such as obesity, inflammatory bowel disease and metabolic disorders [23,24,25]. In the context of the vaginal microbiome, research has shown that an insufficient intake of micronutrients such as vitamins A, C, D, E, β-carotene, folate, and calcium may increase the risk of BV [26,27,28,29]. There is also evidence to suggest that an increased carbohydrate intake may fuel *Lactobacillus* spp. growth within the vagina by increasing the free glycogen levels [26,30]. Glycogen is metabolised to lactic acid by Lactobacilli, which promotes an acidic vaginal pH [30]. However, carbohydrates with a high glycaemic index have also been demonstrated to increase the risk of BV in women, a condition generally associated with a low abundance of *Lactobacillus* spp. [26]. With respect to fats, a study by Neggers et al. of 1521 women found that a high dietary fat intake was also associated with an increased risk of BV and that an increased intake of micronutrients such as vitamin E, folate, and calcium decreased the risk of severe BV by 60% [31]. Any generalisation of this study’s findings should be approached with caution, however, as the sample consisted primarily of lower socioeconomic status African American women, a population already known to be at an increased risk of BV [32].

### 4.3. Exercise and Body Mass Index

Despite the known impacts of exercise and body mass index (BMI) on the gut microbiome [33], few studies have assessed their impacts in the context of the vaginal microbiome. Song et al. [34] examined the impacts of exercise on the vaginal microbiome in 26 college-aged women with participants self-reporting their average exercise intensity as low, moderate, or high. They found that women who participated in higher-intensity exercise were more likely to have higher alpha diversity within their microbiome, akin to CST IV. Despite this, they did not use a validated self-reporting exercise scale; this is a major limitation of this study. Future studies should incorporate validated self-report methods such as the Borg Rating of Perceived Exertion (RPE) [35] to further explore potential associations between exercise and impacts on the vaginal microbiome.

Raglan et al. [36] assessed the vaginal bacterial composition of 67 obese and 42 non-obese women and reported that obese women were more likely to have *Lactobacillus*-depleted vaginal microbiomes and increased alpha diversity, as well as higher local cytokine levels compared to non-obese women. Additionally, they analysed microbiome changes in a subset of obese women (n = 27) undergoing bariatric surgery. Prior to surgery, they found no significant associations between BMI and the vaginal microbiome. However, six months post-surgery, they observed a significant association between lower BMI and a *Lactobacillus*-dominant microbiome. In contrast, Mirmonsef et al. [30] examined the relationship between free glycogen in vaginal fluid and *Lactobacillus* abundance. They found that free glycogen levels in the lumen of the vagina were higher in women with a high BMI (>30). They also noticed that women with a BMI between 25 and 29.9 (overweight) had three times the odds of having >85% *Lactobacillus* abundance (OR 3.11, 95% CI 1.31–7.37). Daubert et al. [37] examined the relationship between BV and BMI among women living with or at risk of HIV and reported that obese post-menopausal women had a significantly lower risk of BV compared with post-menopausal women with a normal BMI (18.5–24.9); however, this relationship was not significant among pre-menopausal women. In contrast to these studies, Brookheart et al. [38] found that BV prevalence was highest among overweight and obese women compared with lean women, even after adjusting for race. Given the conflicting results surrounding BMI and BV prevalence, further studies are warranted in this area.

### 4.4. Stress

Chronic stress stimulates the hypothalamic-pituitary-adrenal (HPA) axis, promoting the release of cortisol from the adrenal cortex. Stress-related vaginal dysbiosis is hypothesised to be caused by increased cortisol levels which suppress immune activity leading to the loss of *Lactobacillus* sp. dominance [39]. Stress in pregnancy is an established risk factor for preterm birth [40,41,42]. Psychosocial stress also increases the risk of BV [39,43,44]. Culhane et al. [43] reported in 454 pregnant women that chronic stress was a significant and independent risk factor for BV status, even after multivariable analysis. Specifically, women in the moderate- and high-stress groups (as determined by the Cohen Perceived Stress Scale) were 2.3 and 2.2 times more likely to have BV than women in the low-stress group, respectively. These findings were corroborated in a non-pregnant cohort by Nansel et al. [44] using data from the Longitudinal Study of Vaginal Flora. Another study comparing chronic stress and BV in pregnant women found that Black women had significantly higher rates of BV compared with white women [45]. Black women were also more likely to be exposed to chronic stressors at personal and community levels than white women which is likely to explain a significant amount of racial disparity in BV prevalence.

### 4.5. Smoking

Research on the impacts of cigarette smoking on the vaginal microbiome has revealed an increased prevalence of bacterial vaginosis in smokers, as well as a greater risk of preterm birth [2,46,47,48,49,50,51]. Payne et al. [52] analysed the vaginal microbiomes of pregnant women for the presence of three target organisms (*Ureaplasma*, *Mycoplasma*, and *Candida* spp.) previously associated with preterm birth and found that smoking significantly increased the odds of detection of all three. Cigarette smoking is also known to have anti-estrogenic effects, which may negatively impact the growth of *Lactobacillus* spp. in the vagina [53]. Westhoff et al. [53] measured the mid-cycle and luteal phase concentrations of estrogens and progestins of 175 reproductive-aged women and observed that smoking was associated with decreased estrogen levels in both phases. Brotman et al. [46] conducted a cross-sectional study in which 20 smokers and non-smokers were recruited and self-collected vaginal swabs for vaginal CST analysis. They reported that 50% of smokers had CST-IV (*Lactobacillus*-depleted) microbiomes in comparison to just 15% of non-smokers. Additionally, smokers had higher vaginal pH and Nugent Gram stain scores indicative of a BV diagnosis than non-smokers. Among their participants, women with CST-IV microbiomes had 25-fold greater odds of being smokers than those with CST-I microbiomes. While this result demonstrates a significant impact of smoking on vaginal *Lactobacillus* titres, it is important to note that the confidence interval for this association was very wide (aOR 25.61, 95% CI: 1.03–636.61), likely due to the very small sample size, therefore additional studies are needed to validate this finding.

A possible explanation for the reduced *Lactobacillus* titres observed in smokers may be the presence of benzopyrene diol epoxide (BPDE), a chemical in cigarette smoke that has been found in the vaginal secretions of smokers [54]. Pavlova et al. [54] analysed BPDE on *Lactobacillus* sp. in vitro and reported a significant increase in phage induction, which may explain the greater odds of *Lactobacillus*-depletion in smokers. Nelson et al. [55] compared the vaginal metabolomes of smokers and non-smokers and found that nicotine and nicotine metabolites cotinine and hydroxycotinine were significantly higher in the vaginal metabolomes of smokers. They also discovered that smokers with CST-IV microbiomes had significantly higher levels of bioamines, which are known to impact the virulence of infective pathogens and contribute to vaginal malodour. This suggests that smoking may precipitate increased malodour and predispose women to vaginal infections.

### 4.6. Age

Across a woman’s life, the vaginal microbiome undergoes substantial modifications due to various stressors, sex hormones and habits. The vaginal pH is neutral or alkaline during childhood, dominated by anaerobic bacteria, Diphtheroids, coagulase-negative Staphylococci*, E. coli* and *Mycoplasma* species [56,57]. The rise in estrogen that occurs during puberty promotes hyperplasia of the vaginal mucosal epithelium and increases the cellular glycogen content [56]. These changes promote a vaginal microbiome that is dominated by *Lactobacillus* sp. in many, however, is also accompanied by an increase in anaerobic species such as *Atopobium* and *Prevotella* [56]. Numerous studies have reported that women of reproductive age typically have microbiomes dominated by one or more *Lactobacillus* sp., or are *Lactobacillus*-depleted [4,56,57]. As women approach menopause, a decline in circulating estrogen causes a shift towards a *Lactobacillus*-depleted microbiome, with a subsequent rise in vaginal pH [56,57]. These findings are consistent with Brotman et al. [58], who found premenopausal women were more likely to have CST I and III microbiomes, whereas post-menopausal women were more likely to have CST IV-A. The vaginal microbiomes of women with mild or moderate vulvovaginal atrophy (VVA), a condition that causes vaginal dryness and soreness in postmenopausal women due to estrogen decline, had 25-fold greater odds of being classified as CST IV-A compared to women with no VVA [58].

Hormone replacement therapy has been associated with the restoration of *Lactobacillus* abundance in postmenopausal women. Several studies have reported that postmenopausal women on hormonal treatment had significantly higher free glycogen levels and increased *Lactobacillus* abundance compared with those not on hormone treatment [57,59,60,61]. Ribeiro et al. [62] compared the effects of isoflavone administration and probiotics to hormonal therapy on 60 postmenopausal women and found that after 16 weeks of treatment, the hormonal therapy group had significantly improved menopausal symptoms, lower vaginal pH and increased *Lactobacillus* sp. abundance. Similarly, Pabich et al. [61] analysed the vaginal communities of 463 post-menopausal women and reported that *Lactobacillus* sp. were present in 62% of women and significantly more prevalent in those receiving hormonal replacement therapy in the previous year.

### 4.7. Menstrual Cycle

Vaginal CSTs are also known to shift during menses, before reverting to their original states later in the menstrual cycle (Figure 1) [13,58]. There is evidence to suggest that menses is accompanied by increases in alpha diversity along with a decrease in the abundance of *Lactobacillus* spp. [34]. Srinivasan et al. [63] reported that the relative abundance of *Gardnerella vaginalis* and *L. iners* increased during menses and was accompanied by reduced quantities of *L. crispatus* and *L. jensenii*. After menses, the relative abundance of *G. vaginalis* and *L. iners* decreased and there were simultaneous increases in the relative abundance of *L. crispatus* and *L. jensenii* [63]. Estrogen levels peak prior to ovulation and in the luteal phase of the menstrual cycle [64]. The luteal phase is more stable in terms of microbial composition which correlates with the higher circulating concentrations of sex hormones such as estrogen and progesterone [58]. Multiple studies have reported consistent findings of *Lactobacillus*-depletion during menses when estrogen levels are lowest, with a shift towards *Lactobacillus*-dominance just prior to ovulation when estrogen levels are highest [34,64,65,66]. This is consistent with the idea that the vaginal microbiome appears to be less stable during menstruation. It is important to note, however, that a recent study by Chaban et al. [15] that followed 27 reproductive-aged women throughout a single menstrual cycle found that the vaginal microbiomes of women remained relatively stable, with minimal variations in diversity and richness. However, only one-quarter of participants provided vaginal samples during menses, which may explain these discrepant results. Future, larger studies following women through a greater number of menstrual cycles are required to validate these findings.

### 4.8. Contraception

Hormonal contraceptives such as the combined oral contraceptive pill (COCP), and the hormonal intrauterine device (IUD) release sustained amounts of estrogen and progestin throughout the menstrual cycle, preventing ovulation and rendering cervical mucus impenetrable by sperm [67]. Barrier contraceptives such as condoms prevent genital contact as well as the transfer of sperm into the vagina, which helps to maintain a healthy vaginal microbiota. In fact, studies have found that condom users have a higher prevalence of H_2_O_2_-producing Lactobacilli [68], and are less likely to exhibit a non-optimal CST III (*L. iners)* microbiome [69]. There is also consistent evidence that hormonal contraceptive use prevents BV [70,71,72,73]. Interestingly, Rezk et al. [74] examined the prevalence of BV, *Trichomonas vaginali*s and *Candida* spp. infections among new users of the COCP or the hormonal IUD and found that the rates of these infections significantly increased after six weeks in both hormonal contraceptive groups but decreased in frequency over time. The increased rate of infection at six weeks may be associated with increased promiscuity and a decrease in condom use after the initiation of hormonal contraception, given that the penile microbiome of a male partner can predict incident BV in women [75]. The protective effects of hormonal IUD use on BV acquisition are yet to be established. Several studies have failed to show any significant protective effects [70,73], and Donders et al. [76] even reported that short-term use of hormonal IUDs increased BV, aerobic vaginitis, and *Candida* spp. rates. However, these rates were reduced back to pre-insertion levels after long-term use; therefore, these results may only be reflective of the brief period of microbial disturbance that likely occurs post-IUD insertion.

### 4.9. Pregnancy

During pregnancy, the vaginal microbiome stabilises and reduces in diversity, generally being dominated by one or two species of *Lactobacillus* spp. [77,78,79]. A longitudinal study by Romero et al. [79] was the first to use 16S rRNA gene sequencing to compare the vaginal microbiomes of pregnant women who delivered at term with those of non-pregnant, healthy women. They reported a statistically significant decrease (95%) in the odds of observing CST IV-B in pregnant women compared with non-pregnant women. Most pregnant women were grouped into CST I and III, whereas non-pregnant women were more likely to have CST III or CST IV-B microbiomes. It was also noted that the vaginal microbiomes of pregnant and non-pregnant women were dynamic and could shift between CSTs, with non-pregnant women more likely to persist in CST IV-B than pregnant women. A similar finding was also reported by MacIntyre et al. [78] who analysed the vaginal microbiomes of 46 British women throughout pregnancy and six weeks postpartum. They found that the vaginal microbiome shifted postpartum to become less *Lactobacillus* dominant with increased alpha diversity and that significant numbers of British women had CST V microbiomes with low alpha diversity. Interestingly, CST IV has been linked to several pregnancy complications [80], which are discussed below. While previous studies have been able to yield statistically significant results when comparing the microbiomes of pregnant and non-pregnant women [77,78,79], they were often performed with low sample sizes of pregnant women and may be influenced by ethnicity [79]. It is important to note that although previous studies have targeted large numbers of African American women [79], MacIntyre et al. [78] reported their observed postpartum changes to the microbiome were independent of ethnicity. Future longitudinal studies which focus on the vaginal microbiomes of different ethnic groups would be beneficial to ensure that these findings are generalisable to the entire population of pregnant women.

## 5. Impact of the Vaginal Microbiome on Health

Vaginal dysbiosis is a non-optimal state whereby the vaginal microbiota is disrupted due to a range of factors such as stress, antibiotics, and sexual activity. A reduction in the relative abundance of protective *Lactobacillus* spp. can increase the vaginal pH and allow colonisation by a range of pathogenic organisms.

### 5.1. Vaginal Infections

Vaginal infections can occur when vaginal dysbiosis allows the overgrowth of opportunistic organisms such as *E. coli*, *G. vaginalis* and bacterial-vaginosis-associated bacteria (BVAB), or when exposed to a range of pathogenic organisms such as *Chlamydia trachomatis* or *Neisseria gonorrhoeae* during sexual activity [81]. Rapid detection and treatment of these infections is crucial as they can predispose women to a range of reproductive health conditions such as preterm birth, PID, and infertility.

### 5.2. Bacterial Vaginosis (BV)

BV is an inflammatory condition caused by vaginal dysbiosis. A BV-associated vaginal microbiome is generally *Lactobacillus*-depleted with an increase in the relative number of anaerobic bacteria such as *G. vaginalis, Prevotella*, and *Mobiluncus* spp. [82]. BV is the most common vaginal infection in reproductive-aged women and is estimated to affect approximately 25% of women globally [83], with an estimated annual global economic burden of USD 4.8 billion [83]. The most frequently observed symptoms of BV include excessive vaginal discharge, fishy odour, vaginal irritation and a vaginal pH greater than 4.5 [82].

The Amsel criteria are currently the gold standard diagnostic method for BV due to their ability to be performed using basic observational and microscopic techniques [84]. A diagnosis of BV is given when three of four parameters are met: (1) the presence of thin, white, homogenous discharge; (2) the presence of clue cells on wet mount microscopy; (3) pH of vaginal fluid over 4.5; (4) a positive ‘whiff’ test for amines [84,85]. The Nugent criteria are an alternative BV diagnostic method previously considered a gold standard for BV diagnosis. The Nugent scoring system relies solely on Gram stain microscopy for diagnosis and examines the abundance of *Lactobacillus* spp. (Gram-positive rods)*,* versus *Gardnerella* spp. and other anaerobic species (Gram variable rods and curved rods), with a score given based on the proportions of bacteria present; higher Gram-positive rod presence leads to a low Nugent score and vice versa [84,86]. Although somewhat accurate, the Nugent scoring system has reduced in popularity due to its time-consuming methodology and the high skill level required for microscopy. However, in recent years a major flaw in the Nugent scoring system has been brought to light in that *L. iners*, the dominant *Lactobacillus* sp. in CST III vaginal microbiomes (one of the most common), frequently stains Gram-negative, resulting in false positives for BV/vaginal dysbiosis and an associated high Nugent score [87].

Black women are more likely to have *Lactobacillus*-depleted microbiomes compared to white women and are also twice as likely to be diagnosed with BV [1,2,19,32]. A BV-associated microbiome closely resembles that of a CST-IV vaginal microbiome which is common in many reproductive-aged women, particularly African Americans [85]. Asymptomatic BV is a controversial diagnosis whereby women exhibit a vaginal microbiome consistent with BV and are treated for this, despite not displaying any BV symptoms [82]. There is little evidence to suggest treatment is warranted outside the presence of symptoms meeting the Amsel criteria, and proper diagnosis is needed to ensure unnecessary antibiotic treatment for BV is not given to asymptomatic women with CST-IV microbiomes.

### 5.3. Candidiasis

Vulvovaginal candidiasis is an opportunistic yeast infection causing vulvovaginitis and can present with symptoms such as itchiness, thick white discharge, and dysuria [88]. It is estimated to affect approximately 70% of women in their lifetime, however, its absolute incidence is unknown as patients do not always present for care due to the availability of over-the-counter treatment. The organism most responsible for infection is *Candida albicans*, which is a commensal fungus that is part of the normal vaginal microbiota in many women. Most women colonised with *C. albicans* do not display any symptoms of infection, but changes in host and behavioural factors can lead to candidiasis [88,89]. Host-related risk factors include estrogen use, pregnancy, immunosuppression, diabetes mellitus, and broad-spectrum antibiotic use [88]. The use of broad-spectrum antibiotics, especially those with high activity against Gram-positive organisms, is often accompanied by candidiasis due to the depletion of protective *Lactobacillus* sp. causing vaginal dysbiosis, which allows opportunistic organisms such as *C. albicans* to invade the mucosal lining of the vagina and incite an inflammatory response. It has also been recognised that the use of the COCP, hormonal IUD, and some sexual, hygiene and clothing habits can predispose women to infection [89].

### 5.4. Urinary Tract Infections (UTIs)

UTIs affect between 40 and 60% of women at least once in their lifetime [90,91]. Women experience UTIs four times more frequently than males [90], and this is thought to be due to their shorter urethra aiding bacterial ascent into the bladder. A loss of protective *Lactobacillus* spp. in the vagina can allow colonisation of opportunistic UTI-associated organisms such as *E. coli*, *Proteus, Klebsiella,* and *Enterococcus* spp. [90]. These bacteria ascend the urethra causing dysuria, frequent urination, and haematuria. Recent sexual intercourse increases the risk of UTIs in women as it can promote the migration of bacteria into the bladder [90]. Recurrent UTIs (rUTIs) are defined as at least three episodes of a UTI in 12 months or two UTI episodes over 6 months [92]. As women age, the prevalence of rUTIs increases, and this is thought to be due to the decline in endogenous estrogen and *Lactobacillus* spp. that occurs during menopause [91,93,94]. Estrogen replacement therapies such as estrogen creams and rings have been shown to reduce the risk of UTIs in postmenopausal women [61,91]. In a study of 463 postmenopausal women, *E. coli* colonisation was more common in women without estrogen replacement and inversely associated with the presence of *Lactobacillus* spp. [61]. Additionally, *Lactobacillus* retention within the vaginal microbiome was associated with topical or systemic estrogen replacement therapy in the previous year.

### 5.5. Sexually Transmitted Infections (STIs)

Disturbance of the normal vaginal microbiota can allow colonisation by pathogenic organisms which cause STIs. Vaginal dysbiosis is consistently associated with STIs including Human Immunodeficiency Virus (HIV), herpes simplex virus type-2 (HSV-2), Human Papilloma Virus (HPV), and *Trichomonas vaginalis* [2,95]. BV has also been linked to an increased risk of STI acquisition [96,97,98,99,100,101,102]. A meta-analysis of 16 cross-sectional studies by Esber et al. [96] reported that the odds of prevalent BV were 60% greater among HSV-2 positive women compared with HSV-2 negative women. In addition, a meta-analysis by Atashili et al. [101] reported a 60% increased risk of HIV acquisition among women with BV. There is also evidence to suggest that specific vaginal CSTs can affect the risk of STI acquisition [99,103]. Brotman et al. [99] found that the highest proportion of HPV-positive samples came from women with ‘non-optimal’ microbiomes such as CST III and IV, with lower proportions of positive samples found for women with CST I and II microbiomes. This suggests that women with CST IV microbiomes are at an increased risk of HPV and a range of other STIs, potentially due to the absence of the acidic environment that is created via lactic acid production when *Lactobacillus* species are dominant. Early detection and treatment of STIs, as well as BV is vitally important considering their links to PID and infertility [104,105,106].

### 5.6. Pelvic Inflammatory Disease (PID)

PID results from an upper genital tract infection causing damage to the endometrium, fallopian tubes, ovaries, and pelvic peritoneum. *C. trachomatis* and *N. gonorrhoeae* are the most common causes of PID [104]; however, various cervical and enteric bacteria plus BVAB have also been implicated [104,107]. Infection of the vaginal epithelium causes damage, allowing the ascension of bacteria from the cervix into the uterus. PID is often misdiagnosed due to its non-specific symptoms such as pelvic pain and tenderness, and this enables the silent spread of infection into the upper genital tract [108]. Regular STI screening in sexually active women under 25 years is crucial for the prevention of PID as a delayed diagnosis can result in inflammatory sequelae leading to ectopic pregnancy, chronic pelvic pain, and infertility [104]. Several studies have suggested that BV can increase the risk of PID in women, although this association remains unclear [106,109,110,111,112,113]. A large, longitudinal, cohort study by Ness et al. [106] reported that having a vaginal microbiome in the highest tertile in terms of BVAB growth increased the risk of PID by two-fold. Additionally, Haggerty et al. [111] reported that several BVAB such as *Atopobium vaginae, Prevotella,* and *Megasphaera* spp. were significantly associated with subsequent PID. Contradictory to this evidence, other studies by Ness et al. [109] have reported no significant findings between BV and PID. Although the prevalence of PID appears to be declining [107], the risk remains high with approximately 20 million new STIs diagnosed each year in the US [114]. To prevent the progression of infection, the possibility of PID needs to be considered in sexually active women presenting with PID-like symptoms to ensure this infection does not go undiagnosed [104].

### 5.7. Complications of Fertility and Pregnancy

Preterm birth (PTB) is the second most common cause of neonatal mortality worldwide, with approximately 15 million births under 37 weeks’ gestation each year [115]. Intrauterine infections makeup approximately one-quarter of spontaneous PTB cases [116], and can occur due to the ascension of vaginal bacteria into the uterus [117]. A *Lactobacillus*-dominant microbiome is considered the hallmark of optimal vaginal health in reproductive-aged women, and a reduction in *L. crispatus* has been associated with spontaneous PTB (sPTB) in several studies [115,118,119,120]. For example, Fettweis et al. [115] examined the vaginal bacterial profiles of 45 preterm and 90 term birth controls and found that women who delivered preterm had significantly lower levels of *L. crispatus* and higher levels of BVAB1, *Sneathia amnii,* and *Prevotella* spp., among others. It has also been recognised that abnormal vaginal microbiota in early pregnancy can predict late miscarriage and early PTB [121]. A case–control study of 49 pregnant women in which 15 delivered preterm, reported that the risk of PTB was higher for women with CST IV microbiomes with abundances of *Gardnerella* or *Ureaplasma* spp. [80]. Despite its ability to produce lactic acid, *L. iners* has also been recovered in high numbers from women with vaginal dysbiosis [87]. In the context of PTB, Petricevic et al. [122] examined the diversity of *Lactobacillus* in a subset of women delivering term vs. preterm and found that *L. iners* alone was detected in 85% of PTBs, but only 16% of term births. Numerous additional studies have documented the relationship between the vaginal microbiome and PTB, most of which are covered in previous thorough reviews on the topic [123,124,125,126].

The vaginal microbiome has also been linked to natural and artificial reproductive success. A Kenyan study by Lokken et al. [127] reported a 17% decline in natural conception in women who had ever had an episode of BV. Moreover, persistent BV reduced the rate of conception by 43%. In the context of artificial reproductive technology (ART), including in vitro fertilisation (IVF) and IVF-intracytoplasmic sperm injection (IVF-ICSI), some studies have shown that the composition of the vaginal microbiome prior to ART may predict pregnancy outcome [128,129,130]. For example, Bernabeu et al. [131] analysed the vaginal samples of 31 women undergoing ART and reported that the presence of *Lactobacillus* spp. was greater in women who achieved a successful pregnancy. Similarly, Koedooder et al. [128] examined the vaginal microbiome composition of 303 women prior to undergoing IVF OR IVF-ICSI and found that women with *Lactobacillus-*depleted microbiomes were less likely to have successful embryo implantation and that the degree of dominance of *L. crispatus* was an important factor in predicting pregnancy. However, in contrast to the aforementioned study [131], microbiomes containing <60% *L. crispatus* or high titres of *L. iners* correlated with better ART outcomes than microbiomes with >60% *L. crispatus,* suggesting that high titres of *Lactobacillus* spp. may not be beneficial in all cases. Additionally, vaginal dysbiosis has been demonstrated to reduce IVF success [129]. Haahr et al. [129] analysed vaginal samples from 130 IVF patients and reported the prevalence of Nugent-BV in 21% of women, and abnormal microbiota in 28% of women. Interestingly, only 9% of women with abnormal microbiota achieved a successful pregnancy. Given the links between vaginal dysbiosis, PTB and reduced fecundability, additional research targeting the risk factors for BV is needed to prevent these devastating obstetric health consequences.

## 6. Impact of Feminine Hygiene Products and Practices on the Vaginal Microbiome

The use of feminine hygiene products and practices by women to cleanse in and around the genital area with the aim of eliminating vaginal discharge and treating STIs is common, particularly for African and Asian women [132]. However, the use of these products and practices has been linked to adverse vaginal health outcomes [133,134,135]. The widespread use of these products and practices highlights the need for increased education among women regarding intimate female hygiene.

### 6.1. Vaginal Douching

Vaginal douching has been associated with BV [136,137,138,139], PID [140,141,142], PTB [143,144], and reduced fertility [145]. Douching is common in one-third of women in the US and remains prevalent in American and African countries [5,146]. Women perform douching for general cleanliness, to prevent or treat odour and infections, and after sexual intercourse and menses [6,147]. There is a multitude of studies surrounding the impact of vaginal douching on genital health, but many are now outdated or present inconsistent results. Many studies have been conducted in Black women only, who are already at increased risk of these adverse health outcomes. Considering the rise in marketed feminine hygiene products, an updated epidemiological review is needed. Table 1, Table 2 and Table 3 summarise studies assessing the association between vaginal douching and adverse health outcomes including BV, vaginal dysbiosis, and PID. In most of the literature, significant associations between vaginal douching and BV have been reported; however, it is important to note that some studies also report no significant associations [148,149,150,151].

### 6.2. Other Specialised Feminine Hygiene Products

Aside from douching, women use a variety of other feminine hygiene products and practices to cleanse in or around the genital area to remove excess sweat, urine, odour, and discharge [7]. A study by Crann et al. [9] found that women who reported the use of any feminine hygiene product or practice had three times the odds of reporting adverse health conditions such as BV, UTIs or STIs. They also reported that participants using feminine washes/gels had 3.5 and 2.5 times the odds of reporting BV and UTIs, respectively. It appears that these products may reduce the relative abundance of *Lactobacillus* species. Fashemi et al. [156] examined the effects of a vaginal moisturiser (Vagisil), lubricant, nonoxynol-9 and douche on *L. crispatus* in vitro. After two hours, nonoxynol-9 and Vagisil had suppressed *Lactobacillus* growth and at 24 h they had a complete bactericidal effect. Additionally, the lubricant had bactericidal effects within 24 h, however, there were no significant effects of the douche on bacterial growth. Sabo et al. [12] studied the association between vaginal washing and vaginal bacterial concentrations in Kenyan and US women. Among US women, vaginal washing was associated with a significantly higher likelihood of detection of BVAB1/2, *A. vaginae*, *G. vaginalis and Megasphaera* spp., among other bacteria. Crann et al. [9] found that many women also use products not marketed for the genital area including hand sanitisers, baby wipes, oils, shaving cream and body lotions. For example, 41.6% and 2.1% had used baby wipes externally and internally, respectively. The impact of these general cleansing products on the sensitive vaginal microbiome is yet to be established and given the use of these products among women, additional studies in this area are warranted.

## 7. Limitations of Previous Studies and Future Directions

Previous studies examining associations between feminine hygiene practices and the vaginal microbiota have limitations. First and foremost, some studies failed to define internal from external product use and often used the word ‘vagina’ as an all-encompassing term for the genital area. The vulva and vagina are two distinctly different areas that comprise different microbial environments [7]. Crann et al. [9] conducted a thorough survey of the prevalence of certain hygiene products and practices in Canadian women, however, did not conduct any bacterial profiling analyses. In contrast, Sabo et al. [12] conducted a thorough bacterial profiling analysis of the vaginal microbiome but failed to include a survey or adequate description of the feminine hygiene products used. In future, studies which incorporate surveys on the use of feminine hygiene products and practices, along with mid-vaginal swabs for bacterial profiling analyses would be beneficial to ascertain the impact of hygiene practices on the vaginal microbiome.

Additional research is needed to help inform women about which vaginal hygiene practices have the potential for negative impacts on vaginal health, however, at present, the use of vaginal douches is well-established as having multiple negative side effects; As such, we recommend women refrain from using these products without first consulting with their general practitioner.

## 8. Conclusions

The vaginal microbiome is a sensitive microenvironment prone to disruption by several factors including menstrual cycle, age, contraception, smoking, and intimate hygiene practices. The use of vaginal douches is not recommended based on their links to vaginal dysbiosis, bacterial vaginosis, and pelvic inflammatory disease. Studies on the impacts of other intimate hygiene products and practices such as feminine wipes, washes and sprays are limited. Future studies analysing the impacts of a broader range of intimate feminine hygiene products on the vaginal microbiome are needed to ascertain the potential benefits and/or consequences of their routine use by women.

## Figures and Tables

**Figure 1 microorganisms-11-00298-f001:**
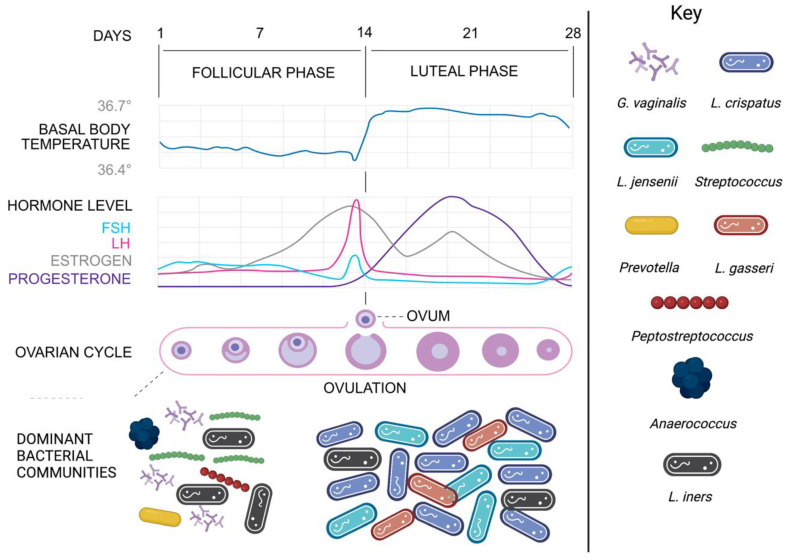
**Bacterial Dynamics of Major Bacterial Genera and Species throughout the Menstrual Cycle.** The Follicular phase (Day 1–7) occurs following menses and is characterised by the gradual increase in levels of follicle-stimulating hormone (FSH), estrogen, and luteinising hormone (LH); the vaginal microbiome at this time is typically dominated by various anaerobes and *L. iners.* During ovulation (~Day 14) and throughout the luteal phase (Day 15–28), levels of circulating estrogen are high, resulting in the dominance of optimal *Lactobacilli* such as *L. crispatus*, *L. gasseri*, and *L. jensenii.* Created with BioRender.com.

**Table 1 microorganisms-11-00298-t001:** Associations between vaginal douching (VD) and vaginal dysbiosis.

Authors (Year)	Racial/Demographic Focus	Study Design	Sample Size	Key Findings
Yıldırım et al. (2020) [6]	Turkish women	Descriptive study	190 women	A significant association reported between VD and history of vaginal infection (*p* < 0.01), as well as women with current vaginal infection and VD. No significant difference reported between VD and non-VD with respect to vaginal microbiota.
Lokken et al. (2019) [152]	Kenyan women	Cross-sectional study	272 women	Vaginal washing in prior week associated with a 44% decrease in *Lactobacillus* detection by culture (aPR 0.56, 95%CI 0.37–0.85). There was a larger reduction in H_2_O_2_-producing *Lactobacillus* with increased washing frequency. (*p* < 0.05)
Baeten et al. (2009) [153]	Kenyan sex workers	Prospective cohort study	1020 women	Vaginal washing (water only or soap plus water) reduced the likelihood of *Lactobacillus* sp. isolation by 40%.
Sabo et al. (2019) [12]	US and Kenyan women	Analysis from Preventing Vaginal Infections (PVI) trial	234 women	US women: Vaginal washing was associated with a higher likelihood of BVAB1 detection (RR 1.55,95%CI 1.15–2.04, *p* = 0.004), BVAB2 (RR 1.99, 95%CI 1.46–2.71, *p* < 0.001) and *G. vaginalis* (RR 1.08, 95%CI 1.01–1.16, *p* = 0.02), among other species. Kenyan women: No association found between vaginal washing and bacterial detection.

**Table 2 microorganisms-11-00298-t002:** Associations between vaginal douching and BV.

Authors (Year)	Racial Focus	Study Design	Sample Size	BV Diagnosis (Amsel/Nugent)	Key Findings
Fonck et al. (2001) [154]	Kenyan female sex workers	Randomised, placebo-controlled trial	543 women	N/A	Douching in general and douching with water and soap associated with BV (*p* = 0.05 and *p* = 0.04, respectively). No significant relationship between douching and risk of STIs/HIV.
Ness et al. 2002 [138]	US women	Cross-sectional study	1200 women	N/A	Douching at least once per month associated with increased frequency of BV. Those douching within 7 days prior were at highest risk (OR 2.1, 95% CI 1.3–3.1). Gonococcal or Chlamydial cervicitis not associated with douching.
Brotman et al. (2008) [139]	US women	Longitudinal study-marginal structural modelling	3620 non-pregnant women	Nugent criteria	Regular douching associated with increased risk of BV compared with no douching (RR 1.21, 95%CI 1.08–1.38).
Klebanoff et al. (2010) [136]	No racial focus	Longitudinal cohort study	3620 women	Nugent criteria	Douching associated with BV (Prevalence Ratio for weekly or greater vs. never 1.17, 95%CI: 1.09–1.26).
Luong et al. (2010) [148]	Canadian women	Nested case–control study	5092 women	Nugent criteria	Vaginal douching was associated with BV (*p* < 0.05) and PTB (*p* < 0.05) in bivariate analysis, but not multivariate analysis.
Brotman et al. (2010) [151]	US women	Cohort study	39 women	Nugent criteria	Vaginal douching practised a day prior to sampling trended towards association with BV, however it was not statistically significant (aOR 3.71, 95% CI 0.79–17.36).
Esber et al. (2016) [150]	Malawi women	Cross-sectional study	200 women	Nugent criteria	95% of women reported use of at least one intravaginal practice (IVP). 51% reported a BV infection. No significant associations between IVP and BV.
Ranjit et al. (2018) [137]	Nepalese women	Descriptive cross-sectional study	160 non-pregnant women	Nugent criteria	Women with daily douching habits more likely to have BV (32.1%) than women who occasionally douched (23.7%) (*p* = 0.015)
Crann et al. (2018) [9]	Canadian women	Cross-sectional survey	1435 women	N/A	Participants who douched in the previous six months had 7 times the odds of reporting BV.

**Table 3 microorganisms-11-00298-t003:** Association between vaginal douching and Pelvic Inflammatory Disease (PID).

Authors (Year)	Racial/Demographic Focus	Study Design	Sample Size	Key Findings
Zhang et al. (1997) [140]	No racial focus	Meta-analysis	N/A	Vaginal douching increased overall risk of PID (RR 1.73, 95%CI 1.07–2.79) and ectopic pregnancy (RR 1.76, 95%CI 1.10–2.82).
Scholes et al. (1993) [141]	US women	Case-control study	131 cases 294 controls	Women who douched in previous 3 months (aOR 2.1, 95%CI 1.2–3.9) and twice per week (OR 3.9, 95%CI 1.4–10.9) had higher risk of PID.
Ness et al. (2005) [106]	US women	Prospective observational study	1199 women	Douching once or twice per month not associated with PID (aHR 0.76, 95%CI 0.42–1.38) nor Gonococcal/Chlamydial infection (aHR 1.16, 95%CI 0.76–1.78).
Shaaban et al. (2013) [142]	Egyptian women	Cross-sectional observational study	620 women	History of PTB was reported in 19.2% of women who douched vs. 11.9% of non-douching women. There was a history of PID in 13.2% of women who douched vs. 6.0% of non-douching women (*p* = 0.008).
Turpin et al. (2021) [155]	US women	Longitudinal study	2956 women	Nugent BV (aHR 1.53, 95% CI 1.05–2.21), Amsel BV (aHR 2.15, 95%CI 1.23–3.75), and vaginal douching (aHR 1.47, 95%CI 1.03–2.09) associated with incident PID.

## Data Availability

No new data were created during the formulation of this review.

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
