# Peer review of "The Vaginal Microbiome in Health and Disease—What Role Do Common Intimate Hygiene Practices Play?"

_microorganisms, 2023, doi:10.3390/microorganisms11020298_

Round 1

Reviewer 1 Report

The manuscript entitled “The vaginal microbiome in health and disease - what role do common intimate hygiene practices play?” discussed the role of feminine hygiene practices, the factors affecting normal vaginal microbiome, the impact of vaginal microbiome, feminine hygiene products, hygiene practices, and vaginal douching on female health and diseases. The authors also discussed the shortcomings of current and previous studies.

The review was well written and summarized different aspects, factors that affect vaginal microbiome and feminine health. However, major and minor points are required to be improved as below:

 Major comments:

This review is lacking suggestions for future hygiene practices/ to minimize the effects of vaginal dysbiosis which can subsequently affect female health. Also, a conclusion part which summarizes the take-home messages to the readers is necessary

 Minor comments:

The authors should describe a clear definition of “bacterial vaginosis”

Line 172-173: please check the font size

Line 286: Please italicize the “G. vaginalis, L. Crispatus, L. Jensenii, Streptococcus, Provetella, L. gasseri, Peptostreptococcus, Anaerococcus, L. iners”

Line 288: please add more explanation in the figure note of the Figure 1

Line 338: which antibiotics? gram-negative depleting antibiotic? gram-positive depleting antibiotics? anaerobic-depleting antibiotics?

Line 408: please change "2" to "two"

Reviewer 2 Report

In this manuscript by Holdcroft et al., the authors have reviewed the current literature about the factors affecting vaginal microbiome. They also summarized the existing studies of different hygiene practices and their impact on the resident vaginal bacteria in women throughout different populations. Disruption of optimal vaginal microbiome is associated with several adverse health and reproductive outcomes in women and in turn, is a huge social and economic burden to the world. The review is a well-written account of the prevailing information and highlights the gap in the current knowledge of the social, biological and environmental factors influencing vaginal bacteria and thereby, its health. The objective narrative about the existing hygiene practices should help in educating the women to make healthier choices in terms of their vaginal health. The review will also be of interest to the researchers in the field of vaginal microbiome.

Minor comments:

-        Typo in line 21

Round 2

Reviewer 1 Report

Thank you for  your reponses

Author Response

No comments to respond to.